# Contrastive Representation Learning for 3D Protein Structures

## Abstract

Learning from 3D protein structures has gained a lot of attention in the fields of protein modeling and structural bioinformatics. Unfortunately, the number of available structures is orders of magnitude lower than the dataset sizes commonly used in computer vision and machine learning. Moreover, this number is reduced even further, when only annotated protein structures can be considered, which makes the training of existing models difficult and prone to overfitting. To address this challenge, we introduce a new representation learning framework for 3D protein structures. Our framework uses unsupervised contrastive learning to learn meaningful representations of protein structures, making use of annotated and unlabeled proteins from the Protein Data Bank. We show, how these representations can be used in the field of structural bioinformatics to directly solve different protein tasks, such as protein function and structural similarity prediction. Moreover, we show how fine-tuned networks, pre-trained with our algorithm, lead to significantly improved task performance.

## 1 Introduction

In recent years, learning on 3D protein structures has gained a lot of attention in the fields of protein modeling and structural bioinformatics. These neural network architectures process the 3D position of the atoms and/or amino acids in 3D space in order to make predictions of unprecedented performance, in tasks ranging from protein design (Ingraham et al., 2019; Strokach et al., 2020; Jing et al., 2021), over protein structure classification (Hermosilla et al., 2021), protein quality assessment (Baldassarre et al., 2020; Derevyanko et al., 2018), and protein function prediction (Amidi et al., 2017; Gligorijevic et al., 2021) – just to name a few. Unfortunately, learning on the structure of proteins leads to a reduced amount of training data, as compared for example to sequence learning, since 3D structures are harder to obtain and thus less prevalent. While the Protein Data Bank (PDB) (Berman et al., 2000) today contains only around $182\,\text{K}$ macromolecular structures, the Pfam database (Mistry et al., 2020) contains $47\,\text{M}$ protein sequences. Naturally, the number of available structures decreases even further when only the structures labeled with a specific property are considered. We refer to these as annotated protein structures. The SIFTS database, for example, contains around $220\,\text{K}$ annotated enzymes from $96\,\text{K}$ different PDB entries, and the SCOPe database contains $226\,\text{K}$ annotated structures. These numbers are orders of magnitude lower than the data set sizes which led to the major breakthroughs in the field of deep learning. ImageNet (Russakovsky et al., 2015), for instance, contains more than $10\,\text{M}$ annotated images. As learning on 3D protein structures cannot benefit from these large amounts of data, model sizes are limited or overfitting might occur.

In order, to take advantage of unlabeled data, researchers have, over the years, designed different algorithms, that are able to learn meaningful representations from such data without labels (Hadsell et al., 2006; Ye et al., 2019; Chen et al., 2020a). In natural language processing, next token prediction or random token masking are commonly used unsupervised training objectives, that are able to learn meaningful word representations useful for different downstream tasks (Peters et al., 2018; Devlin et al., 2019). Recently, such algorithms have been used to learn meaningful protein representations from unlabeled sequences (Alley et al., 2019), or as a pre-trained method for later fine-tuning models on different downstream tasks (Rao et al., 2019). In computer vision recently, contrastive learning has shown great performance on image classification when used to pre-train deep convolutional neural network (CNN) architectures (Chen et al., 2020a;b). This pre-training objective has also been

used in the context of protein sequence representation learning by dividing sequences in amino acid 'patches' (Lu et al., 2020b), or by using data augmentation techniques based on protein evolutionary information (Lu et al., 2020a). Most recently, the contrastive learning framework has been applied to graph convolutional neural networks (You et al., 2020). These techniques were tested on protein spatial neighboring graphs (graphs where edges connect neighbor amino acids in 3D space) for the binary task of classifying a protein as enzyme or not. However, these algorithms were designed for arbitrary graphs and did not take into account the underlying structure of proteins.

In this work, we introduce a contrastive learning framework for representation learning of 3D protein structures. For each unlabeled protein chain, we select random molecular sub-structures during training. We then minimize the cosine distance between the learned representations of the sub-structures sampled from the same protein, while maximizing the cosine distance between representations from different protein chains. This training objective enables us, to pre-train models on all available annotated, but more importantly also unlabeled, protein structures from the PDB. The obtained representation can later be used as a weight initialization strategy to improve performance on different downstream tasks, such as protein structure classification and function prediction. Moreover, we show how the learned protein representation is able to capture protein structural similarity and functionality, by embedding proteins from the same fold or with similar functions close together in this space.

The remainder of this paper is structured as follows. First, we provide a summary of the state-of-the-art in Section 2. Then, we introduce our framework in Section 3. Later, in Section 4, we describe the experiments conducted to evaluate our framework and the representations learned, and lastly, we provide a summary of our findings and possible lines of future research in Section 5.

## 2 RELATED WORK

**3D protein structure learning.** Early work on learning from 3D protein structures used graph kernels and support vector machines to classify enzymes (Borgwardt et al., 2005). Later, the advances in the fields of machine learning and computer vision brought a new set of techniques to the field. Several authors represent the protein tertiary structure as a 3D density map, and process it with a 3D convolutional neural network (3DCNN). Among the problems addressed with this technique, are protein quality assessment (Derevyanko et al., 2018), protein enzyme classification (Amidi et al., 2017), protein-ligand binding affinity (Ragoza et al., 2017), protein binding site prediction (Jiménez et al., 2017) and protein-protein interaction interface prediction (Townshend et al., 2019). Other authors have used graph convolutional neural networks (GCNN) to learn directly from the protein spatial neighboring graph. Some of the tasks solved with these techniques, are protein interface prediction (Fout et al., 2017), function prediction (Gligorijevic et al., 2021), protein quality assessment (Baldassarre et al., 2020), and protein design (Strokach et al., 2020). Recently, several neural network architectures, specifically designed for protein structures, have been proposed to tackle protein design challenges (Ingraham et al., 2019; Jing et al., 2021), or protein fold and function prediction (Hermosilla et al., 2021).

**Protein representation learning.** Protein representation learning based on protein sequences is an active area of research. Early works used similar techniques as the ones used in natural language processing to compute embeddings of groups of neighboring amino acids in a sequence (Asgari & Mofrad, 2015). Recently, other works have used unsupervised learning algorithms from natural language processing such as token masking or next token prediction (Peters et al., 2018; Devlin et al., 2019) to learn representations from protein sequences (Alley et al., 2019; Rao et al., 2019; Min et al., 2020; Strodthoff et al., 2020). This year, Lu et al. (2020b;a) have suggested using contrastive learning on protein sequences, to obtain a meaningful protein representation. Despite the advances in representation learning for protein sequences, representation learning for 3D protein structures mostly has relied on hand-crafted features. La et al. (2009) proposed a method to compute a vector of 3D Zernike descriptors to represent protein surfaces, which later can be used for shape retrieval. Recently, Guzenko et al. (2020) used a similar approach, to compute a vector of 3D Zernike descriptors directly from the 3D density volume, which can be used later for protein shape comparison. The annual shape retrieval contest (SHREC) usually contains a protein shape retrieval track, in which methods are required to determine protein similarity from different protein surfaces (Langenfeld et al., 2019; 2020). Some of the works presented here, make use of 3DCNNs or GCNNs to achieve this goal.

However, they operate on protein surfaces, and are either trained in a supervised fashion on the binary shape similarity task, or pre-trained on a classification task.

**Contrastive learning.** In 1992, Becker & Hinton (1992) suggested training neural networks through the agreement between representations of the same image under different transformations. Later, Hadsell et al. (2006) proposed to learn image representations by minimizing the distance between positive pairs and maximizing the distance between negative pairs (see Figure 1). This idea was used in other works by sampling negative pairs from the mini-batches used during training (Ji et al., 2019; Ye et al., 2019). Recently, Chen et al. (2020a;b) have shown how these methods can improve image classification performance. You et al. (2020) have transferred these ideas to graphs, by proposing four different data transformations to be used during training: node dropping, edge perturbation, attribute masking, and subgraph sampling. These ideas were tested on the commonly used graph benchmark PROTEINS (Borgwardt et al., 2005), composed of only $1,113$ proteins. However, since this data set is composed of spatial neighboring graphs of secondary structures, the proposed data augmentation techniques can generate graphs of unconnected chain sections. In this paper instead, we suggest using a domain-specific transformation strategy, that preserves the local information of protein sequences.

## 3 3D PROTEIN CONTRASTIVE LEARNING

### 3.1 PROTEIN GRAPH

In this work, the protein chain is defined as a graph $\mathcal{G} = (\mathcal{N}, \mathcal{R}, \mathcal{F}, \mathcal{A}, \mathcal{B})$, where each node represents the alpha carbon of an amino acid with its 3D coordinates, $\mathcal{N} \in \mathbb{R}^{n \times 3}$, being $n$ the number of amino acids in the protein. Moreover, for each node, we store a local frame composed of three orthonormal vectors describing the orientation of the amino acid wrt. the protein backbone, $\mathcal{R} \in \mathbb{R}^{n \times 3 \times 3}$. Lastly, each node has also $t$ different associated features with it, $\mathcal{F} \in \mathbb{R}^{n \times t}$. The connectivity of the graph is stored in two different adjacency matrices, $\mathcal{A} \in \mathbb{R}^{n \times n}$ and $\mathcal{B} \in \mathbb{R}^{n \times n}$. Matrix $\mathcal{A}$ is defined as $\mathcal{A}_{ij} = 1$ if amino acids $i$ and $j$ are connected by a peptide bond and $\mathcal{A}_{ij} = 0$ otherwise. Matrix $\mathcal{B}$ is defined as $\mathcal{B}_{ij} = 1$ if amino acids $i$ and $j$ are at a distance smaller than $d$ in 3D space and $\mathcal{B}_{ij} = 0$ otherwise.

### 3.2 CONTRASTIVE LEARNING FRAMEWORK

Inspired by recent works in the computer vision domain (Ye et al., 2019; Ji et al., 2019; Chen et al., 2020a), our framework is trained by maximizing the similarity between representations from sub-structures of the same protein, and minimizing the similarity between sub-structures from different proteins. More formally, given a protein graph $\mathcal{G}$, we sample two sub-structures $\mathcal{G}_i$ and $\mathcal{G}_j$ from it. We then compute the latent representations of these sub-structures, $h_i$ and $h_j$, using a protein graph encoder, $h_i = E(\mathcal{G}_i)$. Based on the findings of Chen et al. (2020a), we further project these latent representations into smaller latent representation, $z_i$ and $z_j$, using a multilayer perceptron (MLP) with a single hidden layer, $z_i = P(h_i)$. Lastly, the similarity between these representa-

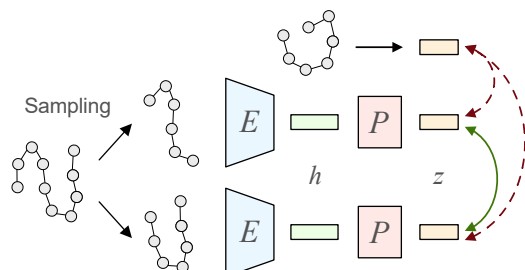

Figure 1: For each protein we sample random substructures which are then encoded into two representations, $h$ and $z$, using encoders $E$ and $P$. Then, we minimize the distance between representations $z$ from the same protein and maximize the distance between representations from different proteins.

tions is computed using the cosine distance, $s(z_i, z_j)$. In order to minimize the similarity between these representations, we use the following loss function for the sub-structure $\mathcal{G}_i$:

$$l_i = -log\frac{exp(s(z_i, z_j)/\tau)}{\sum_{k=1}^{2N} \mathbb{1}_{[k \neq i, k \neq j]} exp(s(z_i, z_k)/\tau)} \qquad (1)$$

where $\tau$ is a temperature parameter used to improve learning from 'hard' examples, $\mathbb{1}_{[k \neq i, k \neq j]} \in [0, 1]$ is a function that evaluates to 1 if $k \neq i$ and $k \neq j$, and $N$ is the number of protein structures in the

current mini-batch. To compute $l_j$ we use again Equation 1, but exchange the role of $i$ and $j$. This loss has been used before in the context of representation learning (Chen et al., 2020a; van den Oord et al., 2018), and as in previous work, our framework does not explicitly sample negative examples but uses instead the rest of sub-structures sampled from different proteins in the mini-batch as negative examples. In the following subsections, we will describe the different components specific to our framework designed to process protein structures.

### 3.3 SUB-STRUCTURE SAMPLING

As Chen et al. (2020a) demonstrated, the data transformation applied to the input, is of key importance to obtain a meaningful representation. Among the different transformations tested, the authors found out that image cropping and adding random noise to the pixel values were the transformations that resulted in more informative representations. In this work, we propose to use a domain-specific cropping strategy of the input data.

Proteins chains are composed of one or several stable sub-structures, called protein domains, which reoccur in different proteins. These sub-structures can indicate evolutionary history between different proteins, as well as the function carried out by the protein (Ponting & Russell, 2002). Our sampling strategy uses the concept of protein sub-structures to sample for each protein two different continuous sub-structures along the polypeptide chain. We achieve that, by first selecting a random amino acid in the protein chain $x_i \in \mathcal{N}$. We then travel along the

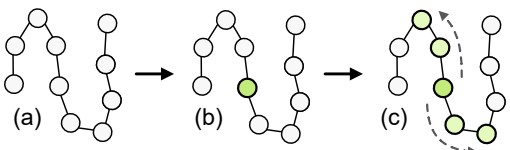

Figure 2: Our sampling strategy used during contrastive learning. For a protein chain **(a)**, we select a random amino acid **(b)**. Then we travel along the chain in both directions until we have a certain percentage $p$ of the sequence covered **(c)**.

protein sequence in both directions using the adjacency matrix $\mathcal{A}$ while selecting each amino acid $x_{i+t}$ and $x_{i-t}$ in the process. This process continues until we have covered a certain percentage $p$ of the protein chain, whereby our experiments indicate that a value of $p$ between $40\%$ and $60\%$ provides the best results (see Section 4). If during this sampling we reach the end of the sequence in one of the directions, we continue sampling in the other direction until we have covered the desired percentage $p$. The selected amino acids compose the sub-structure that is then given as input to the graph encoder $\mathcal{E}$. Figure 2 illustrates this process. Note that, since our framework learns from unlabeled data, we do not sample specifically protein domains from the protein chain, which would require annotations. We instead sample random sub-structures that might be composed of a complete or partial domain, or, in large proteins, even span several domains. The training objective then enforces a similar representation for random sub-structures of the same protein chain, where the properties of the complete structure have to be inferred. We will show in our experiments, that these properties are able to encode similar structural classifications as the ones developed over the years by researchers, as well as information related to the protein function.

### 3.4 PROTEIN ENCODER

The information captured by a learned representation using contrastive learning strongly depends on the network architecture used to encode the input (Tschannen et al., 2020). Therefore, we define the following requirements for our protein encoder. The memory footprint of the network has to be minimal, to allow for large batch sizes (Chen et al., 2020a), all the input data has to be computed on-demand to process our random sub-structures, and the operations must be able to completely differentiate protein structures. We base our protein encoder on the architecture proposed by Hermosilla et al. (2021), whereby we perform several modifications to fulfill our requirements. In the following paragraphs, we describe the proposed network architecture.

**Convolution operation.** Similar to CNNs and GCNNs where a message passing operation is used to update the features of a node, based on the information of the neighboring nodes, Hermosilla et al. (2021) proposed a message-passing operator for 3D protein structures. This operator computed the value of a new feature in a node $x_i$, by aggregating the features from all neighboring nodes $x_j$ at a distance smaller than $d$ in 3D space. The features from neighboring nodes were first scaled by a learned function represented by an MLP with one hidden layer that received as input the edge features

from node pairs $x_i x_j$, resulting in the following operation:

$$F_o^l(\mathcal{G}, x_i) = \sum_{j \in \mathcal{N}(x_i)} F_j^{l-1} k_o(f(\mathcal{G}, x_i, x_j)) \tag{2}$$

where $\mathcal{N}(x_i)$ is the list of nodes at a distance smaller than $d$ from $x_i$, $k_o$ is the learned kernel function, and $f(\mathcal{G}, x_i, x_j)$ is the function that computes the edge information between node $x_i$ and $x_j$.

In the work of Hermosilla et al. (2021), the function $f$ computed three distances between the two nodes: the Euclidean distance, the shortest path along covalent bonds, and the shortest path along covalent and hydrogen bonds. In this work, we use Equation 2 as our message-passing algorithm but our function $f$ computes instead a different set of edge features, similar to the ones used by Ingraham et al. (2019):

- $\vec{t}$: Three values representing the vector $x_j - x_i$ described in the local frame of node $x_i$, $O_i \in \mathcal{R}$, and normalized by distance $d$.

- $r$: Three values representing the orientation information between the local frames $O_i$ and $O_j$. We use the dot product between the three axes instead of other representations such as quaternions (Ingraham et al., 2019) or 6D representations (Zhou et al., 2019) since it improved performance in our experiments (see Section 4).

- $\delta$: One value representing the shortest path along peptide bond between the two nodes $x_i$ and $x_j$, normalized by $\delta_{max}$.

While the features computed by Hermosilla et al. (2021) provided good results in their experiments, they were rotational invariant and they were not able to differentiate chiral protein sub-structures. By incorporating information of direction, $\vec{t}$, and orientation, $r$, in our kernel, the operations become rotation equivariant and provide the network with the tools to differentiate such structures. We denote the combination of these two modifications as *Rot. Eq.* in our experiments. Moreover, Hermosilla et al. (2021) also considered the shortest path along covalent and hydrogen bonds. Calculating such distance on demand can be computationally demanding, which is a limiting factor in our setup, since we sample random sub-structures in each training step. Moreover, the resulting connectivity can depend on the method used to compute the hydrogen bonds, and it can vary for highly disordered structures (Zhang & Sagui, 2015). Therefore, we do not include such bonds and let the network detect the necessary secondary structures directly from the protein structure.

The seven edge features computed by $f$ ($\vec{t}$, $r$, and $\delta$) all have values in the range $[-1, 1]$. Similar to positional encoding (Vaswani et al., 2017), we further augment these inputs by applying the function $g = 1 - 2|x|$, which makes all features contribute to the final value of the kernel $k_o$ even when their values are equal to zero. This feature augmentation results in 14 final input values to $k_o$, the original 7 edges features, plus the transformed ones. We refer to those features as *Add. Input* in our experiments. Lastly, we weigh the final value of $k_o$ by a function $\alpha$ to remove discontinuities at distance $d$, where a small displacement of a neighboring node $x_j$ can make it exit the receptive field. Similar to the cutoff function proposed by Klicpera et al. (2020), the function $\alpha$ smoothly decreases from one to zero at the edge of the receptive field, making the contributions of neighboring nodes disappear as they approach $d$. Our function is defined as $\alpha = (1 - tanh(d_i * 16 - 14))/2$, where $d_i$ is the distance of the neighboring node normalized by $d$. We refer to this function as *Smooth* in our experiments.

**Network architecture.** The network architecture proposed by Hermosilla et al. (2021) processed each protein at the atomic level, and computed features per atom before they were pooled to amino acids. However, processing all atoms of a protein significantly increases computation time and memory footprint. Therefore, in this work, we represent each amino acid by its alpha carbon instead. We use a set ResNet bottleneck blocks (He et al., 2016) and pooling operations as in as Hermosilla et al. (2021) to compute 2048 features for each protein structure. A detailed description of the protein encoder is provided in the supplementary material.

# 4 EXPERIMENTS

In this section, we will describe the experiments conducted to evaluate our methods, and demonstrate the resulting learned representations.

Figure 3: Dimensionality reduction of the protein representations using TSNE (Van der Maaten & Hinton, 2008). **Left:** Proteins from the Fold Classification task, training set on the left, test set on the right, color-coded based on the highest hierarchy level in the SCOPe classification system. **Right:** Proteins from the Enzyme Classification task, training on the left, test set on the right, color-coded based on the highest level of the Enzyme Commission number.

## 4.1 DATA SETS AND TASKS

Our main data set used for unsupervised learning is based on the PDB (Berman et al., 2000). We collected $476, 362$ different protein chains, each composed of at least 25 of amino acids. This set of protein chains was later reduced for each task, removing chains from the set based on the available annotations. All networks were trained for $550 K$ training steps, resulting in 6 days of training.

**Fold Classification.** This data set was first proposed by Hou et al. (2018) which consolidated $16, 712$ proteins of $1, 195$ different folds from the SCOPe 1.75 database (Murzin et al., 1955). Directly from this source, we obtained the 3D structures of these proteins. The data set provides three different test sets with increasing difficulty: Protein, in which proteins of the same family are present in both test and training set; Family, where proteins from the families in the test set are not seen during training; and Superfamily, in which proteins from the same superfamily as the proteins in the test set are not seen during training. Performance is measured with overall accuracy on these test sets. For pre-training, we filtered the PDB data set and removed all annotated protein chains with the same folds as the proteins in the test sets. This procedure generated one PDB data set for each test set composed of proteins with and without annotations. These data sets contain $377, 271$ chains for the Superfamily test set, $313, 616$ chains for the Family test set, and $324, 304$ chains for the Protein test set.

**Enzyme Classification.** This data set was presented by Hermosilla et al. (2021) and it contains $37, 428$ proteins from 384 different EC numbers. The task consists of, given a 3D structure of an enzyme, to predict its complete EC number, e.g. 4.2.3.1, among the 384 numbers available in the data set. The proteins in the data set are split into three sets, training, validation, and testing, whereby proteins in each set do not have more than $50\%$ of sequence similarity with proteins from the other sets. Thus, we obtain $29, 215$ proteins for training, $2, 562$ proteins for validation, and $5, 651$ for testing. Performance is measured with overall accuracy on the test set. For more details, we refer the reader to Hermosilla et al. (2021). For pre-training, we remove all proteins from the PDB data set which belong to the same EC number as the 384 used in the test set, where only the first two digits are considered, e.g. EC4.1. Thus, we obtain $270, 861$ protein chains for pre-training with and without EC labels..

**Protein Similarity.** We used the benchmark proposed by Holm (2019). This data set is composed of 140 protein domains for which similar protein chains have to be found from a set of $15, 211$. Moreover, they provide another set composed of $176, 022$ protein chains that we use to train our distance metric. In this benchmark, different similarity levels are considered based on the SCOPe classification hierarchy, Fold, Superfamily, and Family. Performance is measured with $F_{max}$ (see Appendix D). For pre-training, we filtered the PDB data set and removed all proteins that are annotated with the same Fold as the 140 protein domains. This resulted in $432, 884$ protein chains with and without annotated Fold class.

## 4.2 QUALITATIVE EVALUATION

First, we evaluate the representation learned by mapping the high dimensional space to a 2D representation using TSNE (Van der Maaten & Hinton, 2008). Then, we color-code each point based on the SCOPe and EC number classification schemes (see Figure 3).

Table 1: Performance comparison of the different variants of our pre-trained networks on the Fold and Enzyme classification task. *1-NN* makes predictions by searching the latent space for the nearest neighbor on the training data set, *SVM* indicates results of a support vector machine model trained on the learned protein representations, *MLP* presents the results of an MLP trained using our representations, *Fine-tuning* indicates the performance of a fine-tuned protein encoder, and *No pre-train* indicates the results of a protein encoder trained from scratch.

| | FOLD | | | | ENZYME |
|---|---|---|---|---|---|
| | Super. | Fam. | Protein | Avg | |
| HHSuite | 17.5 % | 69.2 % | 98.6 % | 61.8 % | 82.6 % |
| TMalign | 34.0 % | 65.7 % | 97.5 % | 65.7 % | |
| Kipf & Welling (2017) | 16.8 % | 21.3 % | 82.8 % | 40.3 % | 67.3 % |
| Diehl (2019) | 12.9 % | 16.3 % | 72.5 % | 33.9 % | 57.9 % |
| Derevyanko et al. (2018) | 31.6 % | 45.4 % | 92.5 % | 56.5 % | 78.8 % |
| Gligorijevic et al. (2021) | 15.3 % | 20.6 % | 73.2 % | 36.4 % | 63.3 % |
| Baldassarre et al. (2020) | 23.7 % | 32.5 % | 84.4 % | 46.9 % | 60.8 % |
| Hermosilla et al. (2021) | 45.0 % | 69.7 % | 98.9 % | 71.2 % | 87.2 % |
| Ours (No pre-train) | 47.6 % | 70.2 % | 99.2 % | 72.3 % | 87.2 % |
| Ours (1-NN) | 21.3 % | 47.5 % | 87.7 % | 52.2 % | 46.5 % |
| Ours (SVM) | 32.3 % | 49.2 % | 94.0 % | 58.5 % | 53.5 % |
| Ours (MLP) | 38.6 % | 69.3 % | 98.4 % | 68.8 % | 74.3 % |
| Ours (Fine-tuning) | **50.3** % | **80.6** % | **99.7** % | **76.9** % | **87.6** % |

For the Fold Classification task, we take the training and test set Protein, and color code each data point based on their class according to the SCOPe classification hierarchy. Note, that the model did not see during training any of the folds of the proteins in the test set. We can see, that our representation clusters points from the same class for classes $a$, $b$, $c$, $d$, and $g$. However, points from classes $e$ and $f$ are spread among the other classes.

Moreover, we also use the same evaluation for the Enzyme Classification task. We color code each data point based on the first number from the EC number. Figure 3 shows that, even if the data points do not seem to form a unique cluster for each EC number, data points from small clusters in the embedding all belong to the same EC class. This might be an indication, that the network did not use the higher levels of the EC number classification scheme to cluster data points, but groups proteins based on other properties that are captured beyond the first digit of the EC number.

## 4.3 QUANTITATIVE EVALUATION

In order to quantitatively evaluate the learned representation, we evaluate how the high dimensional space captures human-designed classifications developed over the years. As it is common practice, we use the performance on downstream tasks, to measure the quality of the learned representation.

**Distance-based classification.** For the tasks of Fold and Enzyme Classification, we compute the latent representation $h$ for the training and test sets. We then use the Euclidean distance in the latent space to search for the closest protein in the train set for each protein in the test set. We count a protein classified correctly if the selected closest protein in the training set is from the same class as the queried protein. Table 1 presents the results of this experiment in the row *1-NN*. We can see, that even if the network did not see proteins from those folds during training, it learns a representation that can classify such folds with higher accuracy, than most of the supervised trained methods. However, for the Enzyme classification task, where the network did not

Table 2: Comparison of our distance metric based on the learned representations for the Protein Similarity benchmark.

| | Fold | Super. | Fam. |
|---|---|---|---|
| DaliLite | **0.38** | **0.83** | 0.96 |
| DeepAlign | 0.28 | 0.78 | **0.97** |
| mTMaLign | 0.13 | 0.55 | 0.91 |
| TMaLign | 0.12 | 0.39 | 0.85 |
| Ours (Euclidean) | 0.14 | 0.39 | 0.65 |
| Ours (Cosine) | 0.12 | 0.39 | 0.65 |
| Ours (Cos. Learn) | 0.35 | 0.55 | 0.63 |

see enzymes of the same subclass during pre-training, the performance is lower than other methods but still competitive.

Moreover, we evaluate the learned representation on the Protein Similarity task. Here, we use the Euclidean distance between representations $h$ and cosine distance between representations $z$ as a measure of protein similarity. We can see in Table 2 that, even if our method is not able to outperform state of the art methods such as DaliLite (Holm, 2019) and DeepAlign (Wang et al., 2013), it achieves higher performance in the Fold test set than TMAlign (Zhang & Skolnick, 2005) and mTMAlign (Dong et al., 2018), which are well-established methods to measure protein similarity. For the Superfamily test set it achieves the same performance as TMAlign, but worse than other methods, while in the Family test set it achieves the lowest performance.

Using a distance metric with our learned representations might not be optimal since we do not enforce them to represent a metric space during training. Therefore, we trained an MLP to transform our representation $h$ into a new representation $z'$ which minimizes the cosine distance between proteins of the same class and maximizes it if they are from a different class. We learned three different representations $z'$, to determine if proteins are from the same Fold, Superfamily, or Family. Table 2 presents the results on such experiment. We can see that this learned distance metric achieves the second-best $F_{max}$ for the Fold task, surpasing commonly used methods such as TMAlign (Zhang & Skolnick, 2005), mTMAlign (Dong et al., 2018) and DeepAlign (Wang et al., 2013). For the Superfamily task achieves competitive performance compared to mTMAlign (Dong et al., 2018), whilst for the Family tasks achieves the worst $F_{max}$. This might be an indication that our representation is able to capture the global structure of protein but struggles to capture fine details. When comparing timings for a single query against the $15K$ proteins, our method only takes a few seconds for the query, as it just performs the subtraction/dot product between the representations, plus around four minutes for loading and encoding of the $15K$ proteins. In contrast, DaliLite (Holm, 2019) requires around 15 hours and TMAlign (Zhang & Skolnick, 2005) a bit less than one hour, on a computer equipped with six cores.

**Classifier.** We also measure the quality of the representation, by training different classifiers using such protein representation as input and comparing to other learned and non-learned based baselines. Note that some of these baselines were designed for different tasks. However, this comparison allows positioning our learned representation into context with other available protein encoders. First, we train a support vector machine model (SVM) on our representation for the tasks of Fold and Enzyme classification, row *SVM* in Table 1. As expected, we achieve higher performance than the Euclidean distance method. We also trained an MLP with a single hidden layer on these representations, row *MLP* in Table 1. This method performed better than the SVM in both tasks, and achieved similar performance to the accuracy obtained by training the protein encoder from scratch on the Fold Classification task, row *No pre-train* in Table 1. Furthermore, this classifier outperforms most of the other methods, including the commonly used TMALign method (Zhang & Skolnick, 2005) on the fold classification task, where the TMAlign similarity is used to find for each protein in the test set the most similar protein in the training set.

**Fine tuning.** Lastly, we measure if our pre-trained encoder can be used as a weight initialization scheme in a fine-tuning setup. Table 1 presents the results of this experiment in row *Fine tuning*. We can see, that this setup achieves the highest accuracy in Table 1, surpassing a randomly initialized protein encoder and obtaining new state-of-the-art results on both data sets.

## 4.4 ABLATION STUDIES

In this section, we evaluate the pre-training procedure. First, we analyze how the amount of information removed from the sequence affects the learned representation. We measure it by training an MLP to classify the protein representations according to the Fold classification task. For pre-training, we use the entire PDB data set, and train the protein encoder for $180K$ training steps, which results in two days of computation. From the results in Table 3, we can see that removing between $20\%$ and $40\%$ of the protein chain makes the contrastive objective too easy, and the protein encoder does not learn a rich enough representation. On the other hand, removing between $60\%$ and $80\%$

Table 3: Ablations on the data transformations used during pre-training for the Fold Classification task.

| Transf. | Super. | Fam. | Protein | Avg |
|---|---|---|---|---|
| 60%-80% | 35.2 % | 67.2 % | 98.0 % | 66.8 % |
| 40%-60% | 38.9 % | **70.1** % | **98.8** % | **69.3** % |
| 20%-40% | 32.5 % | 57.0 % | 97.6 % | 62.4 % |
| Sub-graph | **42.1** % | 65.9 % | 98.7 % | 68.9 % |

Table 4: Ablations on the data transformations used on the supervised setting for the Fold Classification task.

| Transf. | Super. | Fam. | Protein | Avg |
|---|---|---|---|---|
| 60%-80% | 9.9 % | 23.8 % | 62.7 % | 32.1 % |
| 40%-60% | 26.7 % | 48.3 % | 91.9 % | 55.6 % |
| 20%-40% | 38.9 % | 63.2 % | 98.0 % | 66.7 % |
| Noise | **47.6** % | **70.2** % | **99.2** % | **72.3** % |

does not preserve enough information, and the performance also suffers. As can be seen, we found that removing between $40\%$ and $60\%$ of the protein chains achieves the best performance.

We further compare our suggested data transformation approach to the graph augmentation technique used by You et al. (2020). You et al. (2020) selected a random sub-graph on the spatial neighboring graph, thus selecting a random area in 3D space. We can see in Table 3, that while You et al. (2020) obtains a good performance on the Superfamily test set, it obtains lower accuracy on the Family and Protein test set. Since the Family and Protein test sets contain proteins with higher sequence similarity to the training set than the Superfamily test set, we hypothesize that our method uses more information of the protein sequence for the representation than the sub-graph method, since the latest sees disconnected sections of the chains during training, while our method always sees a connected sub-chain. Although we acknowledge that both methods can be beneficial for different tasks, we observed that on average our method provides better performance.

Lastly, we evaluate how the proposed data transformation could affect the supervised training of the protein encoder on the Fold Classification task. We can see in Table 4, that in contrast to the unsupervised training, our data transformation technique used as data augmentation reduces performance in this setup. Instead, the best accuracy is obtained by adding a small random Gaussian noise into the 3D coordinates of the alpha carbon. These results align with the ones obtained on other contrastive learning works (Chen et al., 2020a), where these extreme data transformation strategies hurt the supervised training instead of improving its performance.

## 5 CONCLUSIONS

In this paper, we have introduced contrastive learning for protein structures. While learning on protein structures has shown remarkable results, it suffers from rather low availability of annotated data sets, which strengthens the need for unsupervised learning technologies. In this paper, we demonstrated, that by combining protein-aware data transformations with adequately adapted, state-of-the-art learning technologies, we were able to obtain a learned representation without the need for such annotated data. This is highly beneficial, since the availability of annotated 3D structures is limited, as compared to sequence data. Using our learned representation, we could show on relevant protein tasks, that we achieve competitive performance when the distance between representations is used as a metric of protein similarity.

We believe that our work is a first important step in transferring unsupervised learning methods to large-scale protein structure databases. In the future, we foresee, that the learned representation can not only be used, to solve the tasks demonstrated in the paper, but that it can also be helpful, to solve other protein structure problems. Protein-protein interaction prediction, for example, could be addressed using our proposed framework by finding the matching geometric patterns in the learned representations. Additionally, upon acceptance, we plan to release the representations for all PDB proteins, and make our technology available, such that these representations can be updated with newly discovered proteins.

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

Table 5: Ablations on the edge feature elements on the Fold Classification task.

|  | Super. | Fam. | Protein | Avg |
|---|---|---|---|---|
| Baseline | 39.0 % | 65.6 % | 98.6 % | 67.7 % |
| Rot. Eq. | 45.5 % | 69.7 % | 98.9 % | 71.4 % |
| Add. Input | 44.6 % | 67.7 % | 98.7 % | 70.3 % |
| Smooth | 40.7 % | 65.4 % | 98.4 % | 68.2 % |
| Full | **47.6** % | **70.2** % | **99.2** % | **72.3** % |

Table 6: Ablations of the rotation representation used on the Fold Classification task.

|  | Super. | Fam. | Protein | Avg |
|---|---|---|---|---|
| Quat. | 44.0 % | 68.7 % | 98.0 % | 70.2 % |
| 6D | **46.1** % | 68.2 % | 98.7 % | 71.0 % |
| Dot Axis | 45.5 % | **69.7** % | **98.9** % | **71.4** % |

## A  ADDITIONAL ABLATION STUDIES

First, we evaluate the different extensions we incorporated into the convolution operation as proposed by Hermosilla et al. (2021) (see Table 5). Our baseline method uses as edge features the Euclidean distance and the shortest path along the sequence. We evaluate the performance improvement when we substitute the Euclidean distance by direction and orientation information as described in Section 3.4, denoted as *Rot. Eq.* in the table. We also evaluate how transforming the original inputs similar to positional encoding affects the resulting performance, denoted as *Add. Input* in the table. Furthermore, we evaluate the effect of the smoothing function applied towards the boundary of the receptive field, denoted as *Smooth* in Table 5. We can see that adding the components individually to the baseline, results in an improvement of accuracy in all cases. Moreover, when we incorporate all together in our final convolution operation, we experience even a higher improvement, *Full* in Table 5.

Lastly, we evaluated the performance of the model, when changing the representation of the orientation features. Here, we compare Quaternions (Ingraham et al., 2019), with the 6D representation introduced by Zhou et al. (2019), and the simple dot product between the axes of the two frames. Results of this experiment are shown in Table 6. The worse performance is obtained by Quaternions, while the 6D and the dot product obtain similar performance. Although the dot product is not able to represent a full rotation, it obtained a slightly higher performance than the 6D and a faster convergence during training. We hypothesize, that even the 6D representation is more descriptive, it uses more floats than the dot product method wrt. the rest of the inputs to the kernel.

## B  NETWORK ARCHITECTURE

Our neural network receives as input the list of amino acids of the protein. Each protein is then simplified several times with a pooling operation that reduces the number of amino acids by half each step. We use the same pooling operation proposed by Hermosilla et al. (2021) where every two consecutive amino acids are grouped into a new node. The initial features are defined by an embedding of 16 features for each amino acid type that is optimized together with the network parameters. These initial features are then processed by two ResNet bottleneck blocks (He et al., 2016) and then pooled to the next simplified protein representation using average pooling. This process is repeated four times until we obtain a set of features for the last simplified protein graph. The number of features used for each level are $[256, 512, 1024, 2048]$. The radius of the receptive field, $d$, used to compute the adjacency matrix $\mathcal{B}$ in each level are $[8, 12, 16, 20]$ Å. Lastly, in order to obtain a set of features for the complete protein structure we use an order invariant operation that aggregates the features of all nodes. In particular, we use the average of the features of all nodes. Figure 4 provides an illustration of the proposed architecture.

## C  TRAINING PARAMETERS

In this subsection, we will describe the hyperparameters used on the different training setups. All methods were trained on a desktop pc with six cores, 32 Gb of RAM, and a GeForce GTX 2080.

Figure 4: Illustration of our protein encoder. We use an amino acid embedding as our input features that are then processed by consecutive ResNet Bottleneck blocks and pooling operations. To obtain the final protein representation we use the average of the features from the remaining graph nodes.

## C.1 UNSUPERVISED SETUP

To train our models with the contrastive learning objective we used a latent representation $h$ of size 2048 and a projected representation $z$ of size 128. We use Stochastic Gradient Descent (SGD) optimizer with an initial learning rate of 0.3 which was decreased linearly until 0.0001 after a fourth of the total number of training steps. We use a batch size of 256 and a dropout rate of 0.2 for the whole architecture. Moreover, we used a weight decay factor of $1e - 5$.

## C.2 FOLD AND ENZYME CLASSIFICATION

**Supervised:** To train the models on the supervised task of Fold classification we use the SGD optimizer with an initial learning rate of 0.001 which was decreased after 100 epochs to 0.0001 and after 300 epochs to 0.00001. We train the networks for a total of 400 epochs using a batch size of 8. To avoid overfitting we use a dropout rate of 0.2 on the encoder and 0.5 on the final MLP. Moreover, to further regularize the model, we use a weight decay factor of $5e - 4$. We additional augment the data by applying Gaussian noise to the 3D coordinates with a standard deviation of 0.05.

**Fine-tune:** For the fine-tuning setup we use the same parameters and for the supervise method but we use instead a smaller learning rate of 0.0005 which is decreased to 0.00005 after 300 epochs. Moreover, we use linear learning rate warm-up for the first 25 epochs.

## D PERFORMANCE METRIC DALILITE

To evaluate the performance of different methods on the DaliLite benchmark we use $F_{max}$ as defined by Holm (2019). We sort the 15 K proteins based on our distance metric to our target and use the following definition of $F_{max}$:

$$
\begin{aligned}
F_{max} &= \max_n F(n) \\
&= \max_n \frac{2p(n)r(n)}{p(n) + r(n)} \\
&= \max_n \frac{2TP(n)}{n + T}
\end{aligned}
\tag{3}
$$

where $n$ is the rank of the query in the ordered list, i. e. the index of the protein in the sorted list. For the $n$ first results in the ordered list, we define $p(n)$ as the precision, $r(n)$ as the recall, $TP(n)$ as the number of true positives pairs, and $T$ is the number of structures in the class. We compute the final value for the test set by averaging the $F_{max}$ among the 140 test protein domains. For more details on this metric, we refer the reader to Holm (2019).

