# OpenReview forum: "Contrastive Representation Learning for 3D Protein Structures"
_ICLR.cc/2022/Conference — ICLR 2022 Submitted_

### Official Review · Reviewer_c8cK · 2021-10-30

**Correctness:** 4
**Technical Novelty And Significance:** 2
**Empirical Novelty And Significance:** 2
**Recommendation:** 5
**Confidence:** 4

**Main Review:**

positives:
1) The paper addresses an important problem of inferring protein functions and structures using limited number of labeled data points.
2) Using contrastive learning as an unsupervised method for protein embedding seems to be novel.
3) Experimental evolutions are extensive and ablation studies have been conducted.
4) The paper is well-written and understandable.

negatives:
1) Perhaps the main concern is that the results do not justify that the learned embedding has superior advantage over simpler baselines. The embedding unfortunately falls short on the Protein data set in Table 2 compared to baselines. Also in Table 1, applying classifiers (SVM, MLP, and 1-NN) on the learned embedding does not result in a competing classifier suggesting than the learned low dimensional representation does not fully reflect the nonlinearities present in the target classification tasks. Not sure how to justify from these results the embedding is useful beyond an initialization for the Fine-Tuning procedure.
2) The novelty of the method is also limited; the contrastive learning is directly applied to the protein problem and several details of the method including sub-structure sampling and protein encoder are borrowed from Hermosilla et al. and Ingraham et al.

questions:
1) How sensitive are the results of the embedding to the choice of p?
2) Is it possible to train simple classifiers (e.g., SVM, etc.) on the Protein data of Table 2 similar to Table 1?
3) Is there any benefit towards using larger number of proteins in PFAM as the main unsupervised data set? The paper motivates the presence of millions of proteins in PFAM, however, uses couple of hundred thousands as the main data set --- this could enable better results.
4) Can the authors comment on possible interpretability of their embedding regarding features for specific 3D (sub)structures that might emerge from their method? and this give additional edge over the competing methods beyond the quantitative scores?




**Summary Of The Paper:**

The paper proposes an unsupervised method for learning low dimensional representation of proteins borrowing ideas from contrastive learning in the computer vision literature. Experiemnts on three data sets compares the performance of such embedding with baselines algorithms.

**Summary Of The Review:**

Overall while the problem that the paper addresses is important and the method is reasonable, unfortunately, as it stands the proposed embedding method does not yield very convincing results compared to existing baselines in downstream tasks. Added with limited novelty of the method the paper in my opinion is marginally below acceptance; although I remain open to authors discussions in the rebuttal.

---

> ### Author Response · Authors · 2021-11-22
> **Are embeddings useful beyond an initialization for the Fine-Tuning procedure?**
>
> > Not sure how to justify from these results the embedding is useful beyond an initialization for the Fine-Tuning procedure.
>
> Based on the results from Tbl. 1 we believe the representation learned can be used directly for classification. The MLP trained on such representations achieves higher accuracy than all non-learned and learned-based baselines on the Fold classification task only surpassed by Hermosilla et al. 2021. Even if this was not taken into account, if the method is only useful as initialization for a Fine-Tuning procedure, we believe an improvement of 10% on the resulting accuracy over standard initialization is a significant improvement. Lastly, as suggested by the reviewer, we trained an MLP to learn the similarity of proteins and, as Tbl.2 now shows, we achieve competitive performance on this task being orders of magnitude faster than other methods.
>
> ---
>
> > The novelty of the method is also limited.
>
> We respectfully disagree with the reviewer. This is the first representation learning algorithm for 3D protein structures. Moreover, we show how this representation can capture classification schemes derived by humans for many decades in an unsupervised fashion. We believe that even if some of the elements of our framework were proposed before for other tasks, that does not reduce the contributions of this paper.
>
> ---
>
> > How sensitive are the results of the embedding to the choice of p?
>
> Different choices of p are evaluated in Tbl. 3, where we see that increasing or reducing the percentage of protein sequence translates to a decrease in performance.
>
> ---
>
> > Is it possible to train simple classifiers (e.g., SVM, etc.) on the Protein data of Table 2 similar to Table 1?
>
> Following the suggestion of the reviewer, we trained an MLP to transform our latent representation h into a new representation that captures the relation between proteins based on the Fold, Superfamily, and Family classifications using the cosine distance. Tbl.2 shows that our latent representation captures the global structure of proteins and that with a model able to capture the non-linearities of this space we can achieve competitive performance.
>
>
> ---
>
> > Is there any benefit towards using a larger number of proteins in PFAM as the main unsupervised data set?
>
> Unfortunately, we are not able to process protein sequences alone and we need the 3D structures as input. However, as with other learning frameworks, our setup benefits from the amount of data used during training.
>
> ---
>
> > Can the authors comment on possible interpretability of their embedding regarding features for specific 3D (sub)structures that might emerge from their method?
>
> Our embeddings can be used together with gradient-based algorithms commonly used in the field of interpretability of neural networks, such as done by Gligorijevic et al. 2020. These algorithms could explain the concepts behind each individual feature in the latent representation. This could lead to the discovery of new common building blocks in proteins. However, we leave this for future work due to the limited time of the review phase.

---

### Official Review · Reviewer_xGdi · 2021-11-01

**Correctness:** 2
**Technical Novelty And Significance:** 2
**Empirical Novelty And Significance:** 2
**Recommendation:** 3
**Confidence:** 3

**Main Review:**

I found the abstract misleading. It talks about the wider availability of proteins sequence information compared to structural information, leading me to think that the proposed method would learn from both.  But it doesn't: it learns only from 3D sturctural information.

Furthermore, a key missing component of the story is the notion of "annotations."  This concept is mentioned in the abstract ("Moreover, this number is reduced even further, when only annotated protein structures can be considered") but never clearly defined.  A protein can be annotated in many ways, including annotations along the amino acid chains (domain boundaries, protein or DNA binding sites, active sites in enzymes, secondary structural elements, etc.) or annotations of the entire protein sequence (GO terms, KEGG pathways, SCOP annotations).  My best guess is that the authors intend to use the term "annotation" to refer to assignment of protein domains to a protein sequence or structure. ("Note that, since our framework learns from unlabeled data, we do not sample specifically protein domains from the protein chain, which would require annotations.") I do not actually know how protein domains get annotated, but I think it's mostly by sequence similarity.  If this is the case, then it's ironic that the authors are trying to avoid using these annotations during training, since they can presumably be easily generated from 3D structures.

Overall, I found this to be an overly complicated model to test what seems like a straightforward hypothesis (i.e., that this contrastive learning approach is beneficial).  Much of the model seems to derive from a previously published approach (Hermosilla 2021), but with not insignifcant modifications, particularly in the protein encoder.  This approach makes it hard to keep track of what primary claim or claims are being tested here.  In particular, I am not sure if the authors are simply trying to create a state-of-the-art fold recognition engine (in which case the results in Table 2 suggest that they have failed) or simply demonstrate that their contrastive learning approach is beneficial.

The paper needs to be more careful about defining terms before using them. For example, we are told on p. 4 what a protein domain is, but the text refers to protein domains on pp. 1 and 2.

I could not understand what the "Protein" task consists of in the SCOP superfamily setup ("... and Protein, in which proteins of the same family are present during training."): what are the test sequences?

I realize that the enzyme classification task has been presented previously, but I still think it should be described more clearly. I do not understand what a query is in this case, nor whether the train/test split is stratified by class.

I did not find the qualitative evaluations in Section 4.2 to be informative.


**Summary Of The Paper:**

This paper focuses on the general problem of learning a compact embedding of a protein based on its 3D structure.  The key novelty here is to use a contrastive learning procedure: "minimize the cosine distance between the learned representations of the sub-structures sampled from the same protein, while maximizing the cosine distance between representations from different protein chains."

**Summary Of The Review:**

This is a complicated model or set of models, coupled with results that do not clearly support the central hypothesis of the paper, namely, that using contrastive learning outperforms other methods for learning informative embeddigs of protein structures.

---

> ### Author Response · Authors · 2021-11-22
> **Complicated model to test a simple hypothesis**
>
> > This is a complicated model or set of models, coupled with results that do not clearly support the central hypothesis of the paper, namely, that using contrastive learning outperforms other methods for learning informative embeddigs of protein structures.
>
> We respectfully disagree with the reviewer. We can see from Tbl.1 that a pretrained model with our contrastive learning framework leads to an increase of up to 10% in accuracy for some of the tested tasks.
>
> ---
>
> > Abstract is misleading when talking about protein sequences.
>
> We acknowledge that this comparison could be misleading. We have removed this reference from the abstract.
>
> ---
>
> > The concept of annotations is mentioned in the abstract but never clearly defined.
>
> We have properly defined annotation in the text.
>
> ---
>
> > Protein domains are annotated mostly by sequence similarity, which makes obtaining the annotation easily.
>
> If that was the case, the task of finding remote homologs would be solved and it is instead an active area of research (Rao et al. 2019, Strodthoff et al. 2020, or Elnaggar et al. 2020). The annotation process in the SCOPe database (https://scop.berkeley.edu/help/ver=2.08) relies on protein similarity for automated annotations only for certain conditions, and the database is manually curated to remove errors from the automation process. Tbl.1 reports the performance on three different data sets with increased difficulty. The Superfamily test set contains proteins with very low sequence similarity to the training set. We can see that a purely sequence method such as HHSuite obtains a low performance on this task (17.5%) while our method surpasses this accuracy by more than 30 points.
>
> ---
>
> > Complicated model to test what seems like a straightforward hypothesis (i.e., that this contrastive learning approach is beneficial).
>
> Tschannen et al. 2020, demonstrated that the quality of the learned representation strongly depends on the encoder used to encode the data. Therefore, to measure the quality of the contrastive learning approach, we require a state-of-the-art encoder.
>
> ---
>
> > I am not sure if the authors are simply trying to create a state-of-the-art fold recognition engine or simply demonstrate that their contrastive learning approach is beneficial.
>
> As is indicated in the abstract and introduction we present a framework for contrastive learning on 3D protein structures. However, we need a state-of-the-art encoder to achieve the best possible and more meaningful representation possible. If our encoder does not have the tools to differentiate between two different protein structures, the representation learned with the contrastive learning algorithm will not capture these differences and could not be used for downstream tasks. Therefore, we ensured that our proposed protein encoder was able to achieve state-of-the-art results on the selected downstream tasks.
>
> ---
>
> > The paper needs to be more careful about defining terms before using them.
>
> We have modified the paper including these definitions before using them.
>
> ---
>
> > "Protein" task in the SCOP superfamily setup is confusing.
>
> Superfamily is a classification level in the hierarchy of the SCOPe database, just under the Fold level. In this setup, the proteins in the test set are from different superfamilies than all the proteins in the training set. We have modified the text to make this point clearer.
>
> ---
>
> > Enzyme classification task should be described more clearly.
>
> We have included more details of this task in the paper.

---

> > ### Comment · Reviewer_xGdi · 2021-11-29
> > **Remote homology detection**
> >
> > Regarding this comment, "If that was the case, the task of finding remote homologs would be solved and it is instead an active area of research," the task of remote homology detection takes as input protein sequences and uses labels derived from protein structure. In this paper, the authors are taking protein structure as input.

---

> > > ### Author Response · Authors · 2021-11-29
> > > **Remote homology detection**
> > >
> > > We referred the reviewer to the task of remote homology detection since the detection of domains based on sequence similarity, as the reviewer suggested, is by comparing sequences to homologous templates (Wang et al., Protein domain identification methods and online resources). However, these methods fail when no similar templates are available. Therefore, if we want to incorporate this knowledge into our contrastive learning algorithm we would rely only on well-known structures, and new structures or new protein domains will not be used during the pre-training step. We hope this clarified our previous statement.

---

### Official Review · Reviewer_ncDp · 2021-11-03

**Correctness:** 4
**Technical Novelty And Significance:** 3
**Empirical Novelty And Significance:** 2
**Recommendation:** 6
**Confidence:** 4

**Main Review:**

Strengths:
1. The work tries to address an important proteomics problem
2. The writing is generally clear and succinct, but can be improved in some parts (see comments).
3. The approach is interesting and seems like a novel application of existing contrastive learning approaches, adapted for proteins
4. There is a reasonable analysis of the learned models

Weaknesses:
1. Their approach of using contrastive learning seems ill-fitted to learn precise representations of protein structures. They consider sub-chains of proteins as examples. But substructures of the same protein can look very different and have very different functionality? Does it make sense to minimize the cosine distance between them? for instance: an alpha helix and a beta sheet from the same protein can be quite different, so if the cosine distance between these is minimized, it is not clear what the model is learning
2. Can they restrict the similarity constraints within a single domain from a protein, maybe?
3. The evaluation is weak and is done on very coarse labels. For instance, there are currently 1549 different folds (classes) in SCOP and the paper evaluates on 7 high-level folds (classes) that are very different. Many computational papers show results on at least 27 folds (Ding and Dubchak, 2001) or a few hundred folds.
4. Some of the baselines that they compare against were developed for totally different problems. For ex: Baldassarre was for Protein Quality assessment and is specifically tuned for that problem and seems ill-suited for fold classification?
5. The paper's goal is to use unlabeled protein structures that other supervised approaches can't use, but in Section 4.1 they say "chains are removed based on annotations". So, do they only use annotated structures?
6. Table-2 shows that their approach does not work well on protein structural similarity prediction.

Other comments:
------------------------
1. How would embeddings from other unsupervised representation learning approaches like TAPE, or maybe AlphaFold compare?
2. In Table-2, TMalign has either similar or better performance. But in Table-1, TMalign does much worse on fold classification. This seems quite odd. How is it being used for both the tasks?
3. Section 4.3: why does the method not do as well on unseen proteins subclasses for enzyme classification?
4. Section 4.3, computational time comparisons -- Can you compare the training time taken by your approach to the setup for DaliLite and TMAlign that don't need gpus and can be parallelized easily on multicore machines to make them faster.
5. Which data was used for reaching the 20-40%, 40-60% etc, in Section 4.4? Was it a validation set? If the test set is used to check the performance, then this will overfit for the particular task chosen.
6. The "network architecture" section needs better writing to convey the ideas. It is very confusing. Elaborate using a diagram or flow chart? It's not clear how many features and what all is done to the input.
7. How is representation at protein level obtained? Average over all amino acids? And for each amino-acid, is it 16 dimensional? So protein representation is also 16 dimensional?
8. Why not just sample p% starting from a given amino acid? What's the point of getting a window left and right of a specific amino acid?
9. What are the strengths of the
10. Intro in Section 3.3, ""noise" and cropping seem irrelevant to the approach being used.:,


**Summary Of The Paper:**

This paper presents an unsupervised deep learning approach for learning representations of 3D protein structures. They use an objective function motivated by the recent contrastive learning approaches (from computer vision). They show the utility of their model in two downstream applications: protein fold classification, enzyme classification and protein similarity prediction. The main contributions are:
1. they show how the contrastive learning framework can be used in the context of protein structures
2. they analyze the learned representations
3. they show the utility of the model to downstream applications
4. they show how fine-tuning their model leads to an improved performance

**Summary Of The Review:**

I recommend 'weak accept' due to the concerns raised in the above section.

---

> ### Author Response · Authors · 2021-11-22
> **Dissimilar substructures and clarification on evaluation**
>
> >Substructures of the same protein can look very different and have very different functionality.
>
> Whilst this is true, our sampling strategy samples substructures of between 40% and 60% of the original length of the protein sequence. The task for the network then becomes to predict the properties of the whole protein from half of the protein. We can see from our ablation studies (Tbl. 3), that when we decrease the size of the sampled substructure (20%-40%) the performance decreases significantly, validating the comment of the reviewer.
>
> ---
>
> >Can they restrict the similarity constraints within a single domain from a protein, maybe?
>
> Yes, this could be done and possibly could lead to better performance for certain tasks. However, this setup will require using manually annotated data to determine the different domains within the protein.
>
> ---
>
> >The evaluation is weak and is done on very coarse labels.
>
> We regret that we were not able to properly communicate our evaluation setup. The quantitative evaluation was performed on fine-grained labels. The Fold classification task in Tbl. 1 aims at classifying proteins in 1195 different folds. The Enzyme task in Tbl. 1 classifies enzymes in 384 different complete EC numbers.
>
> ---
>
> >Some of the baselines that they compare against were developed for totally different problems.
>
> Thank you for pointing this out. To avoid confusion, we have made the task of each method explicit in the text.
>
> ---
>
> >Methods are trained only with annotated structures?
>
> Our main data set for unsupervised training is the complete PDB database. To have a fair evaluation we removed all proteins from the Fold used in the different test sets. This resulted in a reduced PDB data set, where unannotated proteins and protein annotated on Folds different from the test set remained.
>
> ---
>
> >Table-2 shows that their approach does not work well on protein structural similarity prediction.
>
> Following a suggestion of a reviewer, we learned the distance metric from our embeddings to measure similarity based on Fold, Superfamily, and Family and we obtained competitive performance being orders of magnitude faster than other methods.
>
> ---
>
> >How would embeddings from other unsupervised representation learning approaches like TAPE, or maybe AlphaFold compare?
>
> The embeddings from these methods are derived only from the sequence of the protein and do not use 3D structure to derive such representation. Therefore, our method provides information on the protein that such embeddings are not able to provide.
>
> ---
>
> >Different results comparing TM-Align in Table 2 and Table 1.
>
> Tbl.1 and Tbl.2 report different metrics. While Tbl.1 reports classification accuracy, Tbl.2 reports F_max for protein similarity. In Tbl.1, for each protein in the test set, we use the TMAlign similarity to search the closest protein in the training set to predict the fold of the protein. In Tbl.2, we use the TMAlign similarity to compute the F_max as described in the Appendix. We have modified the text in the paper and made this difference more clear.
>
> ---
>
> >Why does the method not do as well on unseen proteins subclasses for enzyme classification?
>
> In both tasks, the Folds from the test set and EC classes of the test set are removed from the unsupervised training phase. The low performance in the Enzyme task is due to the size of the remaining data set used for pretraining. While for the Fold tasks we use around 325K proteins, for the Enzyme task we only 270K protein remain.
>
> ---
>
> >Can you compare the training time with DaliLite and TMAlign?
>
> While we could compare those times, it can be misleading and lead to misinterpretations. Our networks require around 7 days for pretraining. However, this needs to be done once. Then the representations of each protein can be used for many queries (different tasks) in seconds. DaliLite and TMAlign on the other hand, require hours for each new query. The time required by our method for pre-training could be compared to the time required to build the acceleration hierarchical structures used later by DaliLite.
>
> ---
>
> >Which data was used for reaching the 20-40%, 40-60% etc, in Section 4.4?
>
> For these ablation studies, we use the entire PDB database.
>
> ---
>
> >The "network architecture" section needs better writing to convey the ideas.
>
> We have rewritten the section and made the key points more clear. Due to space limitations, we move the detailed explanation with the requested illustration into the Appendix.
>
> ---
>
> >How is representation at protein level obtained?
>
> After the last layer, we use a symmetric operation to reduce the number of amino acids to a single set of features. In our implementation, we use the average of the features of the last layer. We have stressed this point in the new Appendix.
>
> ---
>
> >Why not just sample p% starting from a given amino acid?
>
> This could also be a valid option resulting in similar probabilities of sampling a given amino acid in the substructure.

---

### Official Review · Reviewer_Ayvu · 2021-11-03

**Correctness:** 3
**Technical Novelty And Significance:** 3
**Empirical Novelty And Significance:** 3
**Recommendation:** 6
**Confidence:** 4

**Main Review:**

Representation learning for 3D protein structures is a broadly relevant problem. The authors proposed using contrastive learning with sub-structure sampling as the data transformation. The empirical results on fold classification and enzyme classification show that the proposed contrastive learning method is able to meaningful representations that capture certain properties of protein structures. The paper is clear and includes detailed ablation studies.

One major concern is that the learned representation from the proposed approach might only capture "global" properties of protein structures and might not be able to capture finer-grain features of protein structures. This might be baked in from the choice of using subgraph sampling as the data transformation. For example, if two antibodies could be structurally very similar to each other but bind to completely different antigens, and it is likely that the learned representation here won't be able to distinguish two structurally similar antibodies. Extending the method to capture finer-grain features in protein 3D structures could enable more use cases for the learned representation.

While learned distance metrics on the protein structure representation space is an important research direction, likely how the distance should be defined would depend on the downstream application. It would be valuable for the authors to clarify when the proposed distance metric in this paper would or would not be a good fit.

While the evaluation tasks (enzyme classification and fold classification) are clearly defined, they are not particularly well motivated and it would strengthen the paper to further explain the biological/clinical significance of these tasks.

The conclusions towards the end of the paper are not well-supported by the empirical results. For example, it is unclear how such a representation would help with solving protein-protein interaction related tasks. More context would be appreciated.

Question: How are variable lengths handled in contrastive learning? Could it be that the representation is implicitly using length to classify fold/EC while baselines such as hhsuite or TMalign cannot make use of the length information?


**Summary Of The Paper:**

This paper studies unsupervised contrastive learning on protein structures, using sub-structure sampling as the data transformation strategy in contrastive learning. The protein structure representation from contrastive learning is then evaluated for three tasks: fold classification, enzyme classification, and protein similarity.

The main contributions are:
1) Demonstrating that contrastive learning with sub-structure sampling is a viable strategy for learning protein structure representations.
2) Improving empirical results for fold classification and enzyme classification when compared to existing methods.

**Summary Of The Review:**

Interesting approach. One major concern is that the learned representation from the proposed approach might only capture "global" properties of protein structures and might not be able to capture finer-grain features of protein structures. Proposed approach shows improvements on fold and enzyme classification tasks, but these tasks may have only limited practical utility.

---

> ### Public Comment · ~Can_Chen3 · 2021-11-13
> **Questions on the Evaluation**
>
> Hello authors,
>
> Very interested in your work!
>
> I have a question. Given known 3D structures, what's the meaning of the fold classification task and the protein similarity task? The answer to the two tasks seems to be determined once the structure is known.
>
> Looking forward to your reply!

---

> > ### Author Response · Authors · 2021-11-22
> > **Know 3D structure**
> >
> > Thanks for showing interest in our work. In our tasks, the 3D structure is known and we aim to automatically classify this new protein into know Folds and measure its similarity to known structures.

---

> ### Author Response · Authors · 2021-11-22
> **Global vs local structure and protein variable length**
>
> >Does the proposed approach only capture "global" properties of protein structures?
>
> The goal of our framework is to obtain a descriptor for a complete protein. Our protein encoder, as the CNN encoders used in images, has a hierarchical architecture that allows it to detect features at different scales and detect local features. Therefore, our encoder possesses the tools to detect local features. If the learned representation can differentiate between similar proteins, as the example suggested by the reviewer, will depend if during the representation learning steps the network saw those two proteins and tried to maximize the cosine distance between them. Based on new experiments, we believe that the current representation might capture a better global structure than local. However, this could be improved in the future by sampling more similar proteins during each training step based on sequence similarity or another metric.
>
> ---
>
> >When the proposed distance metric in this paper would or would not be a good fit?
>
> The distance metrics on the paper, such as Euclidean or cosine, are only used as validation for our latent representation. However, a distance vector D on a region of the latent space X could not measure the same “amount” of similarity as the same vector in region Y. During training we do not enforce our latent space to be a metric space. When we train an MLP to perform classification tasks, it can outperform the Euclidean distance thanks to the non-linearities of the model. Moreover, as suggested by a reviewer, we learned this distance metric using an MLP and the performance on Tbl.2 increased significantly. Therefore, we believe our latent representation might be better used together with an MLP to solve any tasks that require analyzing the global 3D structure of a protein.
>
> ---
>
> >More context on future research directions (Protein-protein interaction).
>
> Determining the interaction between proteins can be seen as a matching problem between certain parts of proteins. Our latent representation encodes the 3D structure of a protein into a latent representation. Therefore, we could train a neural network to find these matching patterns between two latent representations. We have extended such discussion in the paper.
>
> ---
>
> >How are variable lengths handled in the framework?
>
> Our protein encoder can handle variable protein lengths. For each amino acid we use its surroundings to detect certain patterns/features. Then the number of amino acids is divided by two and the process is repeated, increasing the neighborhood at the same time. For a protein of length N, we would require log(N) layers to reduce it to a single amino acid. Therefore, each protein will require a different number of layers and this could lead the network to infer the distance of the sequence. For computational reasons, we instead fix the number of layers to 4 and, after the last one, we use a symmetric operation to reduce the number of amino acids to a single set of features. In our implementation, we use the average of the features of the last layer. Therefore, our network does not receive explicit information on the length of the sequence.
>
> ---

---

### Author Response · Authors · 2021-11-22
**Review overview**

First, we would like to thank all the reviewers for their valuable comments and suggestions. We have uploaded a revised version of the paper addressing most of their concerns. In this new revised version, we have moved some ablation studies into the Appendix to include some of the suggestions of the reviewers. Moreover, we answered each reviewer separately summarizing each comment into a sentence followed by the response.

Sincerely,

The authors

---

### Decision · Program_Chairs · 2022-01-20

**Decision:**

Reject

**Comment:**

In this paper they adapt unsupervised contrastive learning to the problem of representation learning for proteins from 3D structure, using sub-structure sampling for the data transformation. The reviewers have concerns that the application tasks used for evaluation are not particularly impactful tasks, and that additionally, they are likely to not require protein representations that require more nuanced information. There are also concerns about the clarity of the manuscript, and novelty of the technical approach.